# Prenatal Detection of Novel Compound Heterozygous Splice Site Variants of the *KIAA0825* Gene in a Fetus with Postaxial Polydactyly Type A

**DOI:** 10.3390/genes13071230

**Published:** 2022-07-11

**Authors:** Yanyi Yao, Shan Deng, Feng Zhu

**Affiliations:** 1Medical Genetic Center, Maternal and Child Health Hospital of Hubei Province, Tongji Medical College, Huazhong University of Science and Technology, Wuhan 430070, China; yanyiyao@163.com; 2Department of Cardiology, Union Hospital, Tongji Medical College, Huazhong University of Science and Technology, Wuhan 430022, China; dengshan1020@hust.edu.cn

**Keywords:** postaxial polydactyly type A, *KIAA0825*, limb anomaly, whole-exome sequencing, aberrant splicing, functional study

## Abstract

Postaxial polydactyly (PAP) is a common abnormality characterized by extra digits on hands and/or feet. To date, sequence variants in seven genes have been identified in non-syndromic PAP. In the present study, a fetus manifesting non-syndromic postaxial polydactyly type A (PAPA) was found by fetal ultrasonography. To better evaluate fetal prognosis, SNP array analysis and trio whole-exome sequencing (trio-WES) were performed to identify the underlying etiology. Although SNP array analysis revealed no abnormality, trio-WES identified compound heterozygous splice site variants in *KIAA0825*, c.-1-2A>T and c.2247-2A>G in intron 2 and intron 12, respectively. These two splice site variants were absent in control databases and were predicted to influence splicing by in silico analysis. To confirm the potential pathogenicity of the variants, in vitro splicing assays using minigene and RNA from peripheral leukocytes of the heterozygous parents were conducted. Minigene and RT-PCR assays demonstrated that the c.-1-2A>T variant led to the loss of the initiation codon, and the c.2247-2A>G variant mainly resulted in exon 13 skipping. Prenatal WES and subsequent functional studies are important approaches for defining the genetic etiology of fetuses with PAPA and are also essential for accurate genetic counseling and decision making. Taken together, this study expands the spectrum of *KIAA0825* variations in PAPA patients and increases the knowledge of the molecular consequences of *KIAA0825* splice site variants.

## 1. Introduction

Polydactyly, which refers to the occurrence of extra digit(s), is the most frequently observed congenital limb anomaly at birth [1,2]. The prevalence of polydactyly varies from 5 to 19/10,000 live births in different ethnic groups [3]. Polydactyly can occur as an isolated condition or as a part of a more complicated developmental syndrome. In syndromic form, the polydactyly has been found to be a part of 290 well-characterized syndromic malformations [4]. The non-syndromic form of polydactyly is much more common than syndromic polydactyly. Non-syndromic polydactyly, depending on the anatomical location of the extra digits, is classified as preaxial (on the thumb side of the hands and greater toe of the feet), postaxial (on the little finger side of the hands and the fifth toe of the feet) and central polydactyly (involving the middle digits in hands and feet). Clinical studies in Pakistanis showed that central polydactyly is rare and 52% of non-syndromic polydactylies are postaxial polydactylies (PAP) [5].

Non-syndromic PAP is further subclassified into type A (PAPA), with a fully developed extra fifth digit, and type B (PAPB), with an incompletely developed extra digit. According to the different inheritance patterns and mutant loci, PAPA was divided into eleven isoforms with four autosomal dominant loci (PAPA 1–4) and seven autosomal recessive loci (PAPA 5–11). In humans, to date, seven disease-causing genes have been reported in PAPA, including *GLI3* (OMIM: 174,200), *ZNF141*(OMIM: 194,648), *IQCE* (OMIM: 617,631), *GLI1* (OMIM: 165,220), *FAM9**2A* (OMIM: 617,273), *KIAA0825* (OMIM: 617,266) and *DACH1* (OMIM: 603,803), which were associated with PAPA1, PAPA6, PAPA7, PAPA8, PAPA9, PAPA10 and PAPA11, respectively [6,7,8,9,10,11,12].

*KIAA0825* was first identified in 2019 [9]. To date, only four homozygous disease-causing variants of *KIAA0825* have been reported, all in Pakistani consanguineous families, including two frameshift variants [p.(Gln198Thrfs*21); p.(Cys48Serfs*28)], one nonsense variant [p.(Lys725*)] and one missense variant [p.(Leu17Ser)] [3,9,13].

Here, we describe a fetus with two novel splice site variants manifesting non-syndromic PAPA in a Chinese family. To the best of our knowledge, this is the first study of splice site variants of *KIAA0825* in patients with PAPA. This study expands the spectrum of *KIAA0825* variations in PAPA patients and increases knowledge about the biological role of the *KIAA0825* splice site variants, c.-1-2A>T and c.2247-2A>G.

## 2. Materials and Methods

The present study was performed following the Declaration of Helsinki protocols and approved by the Ethics Committee of Maternal and Child Health Hospital of Hubei Province, Tongji Medical College, Huazhong University of Science and Technology, Wuhan, China. Written informed consent for conducting the study and for the publication of photographs was obtained from the parents.

### 2.1. Case Presentation

A 24-year-old primigravida Chinese woman was referred to the Medical Genetics Center at 24 + 3 weeks of gestation due to an abnormal ultrasound examination. The woman and her husband were both healthy and non-consanguineous. None of them had a family history of skeletal anomalies (Figure 1a). The nuchal translucency at 12 weeks of gestation was normal (NT = 1.8 mm), and the second trimester serum screening for trisomy 21, 18 and NTD was low risk. However, at 23 weeks of gestation, a prenatal ultrasound examination revealed that the fetus had PAPA (Figure 1b–e). No other structural abnormalities were found. Amniocentesis was performed at 24 weeks of gestation for SNP array analysis and trio whole-exome sequencing (trio-WES).

### 2.2. SNP Array Analysis

Genomic DNA was extracted from ten milliliters of uncultured amniotic fluid cells by Qiagen DNA Blood Mini Kit following the manufacturer’s protocol. SNP array analysis was performed with CytoScan 750 K array (Affymetrix, Santa Clara, CA, USA), including 550,000 CNVs probes and 200,000 SNP probes, according to the manufacturer’s instructions. The thresholds for our detection criteria for CNVs were set at ≥200 kb for gains, ≥100 kb for losses and ≥10 Mb for the loss of heterozygosity (LOH).

### 2.3. Prenatal Trio-WES 

DNA sample of the fetus was obtained from ten milliliters of uncultured amniotic fluid cells. Parental DNA was extracted from two milliliters of peripheral blood. WES was performed on the fetus–parental trio using the xGen**^®^** Exome Research Panel v1.0 (IDT, Coralville, IA, USA) on the Illumina NovaSeq6000. Exome enrichment of the genomic DNA library was performed according to the manufacturer’s protocols.

Sequencing data were annotated according to the previously described pipeline [14]. Based on the variant annotations, a series of filtering strategies were applied to identify candidate variants from the fetal WES data associated with the phenotype of polydactyly. The filtering steps were as follows: (1) excluding the variants outside exonic and splicing regions (eight bases flanking the exonic boundaries); (2) excluding the variants with minor allele frequency(MAF) ≥0.01 according to public databases (Genome Aggregation Database, 1000 Genomes and Exome Aggregation Consortium database); (3) excluding synonymous variants in exome; and (4) only including de novo variants, homozygous variants, compound heterozygous and hemizygous variant located in the X chromosome inherited from the mother. Then, the remaining variants generated a list of candidate genes. To prioritize the most likely candidate disease-causing gene, all candidate genes were then ranked by Phenolyzer [15]. Finally, the variants in the candidate genes were verified by Sanger sequencing and classified according to ACMG guidelines [16].

### 2.4. Splicing Prediction

Multiple online in silico splice site prediction software were used to evaluate the potential pathogenicity of the splice site variants, including Human Splicing Finder (http://umd.be/Redirect.html, accessed on 15 January 2022), Splice AI (https://spliceailookup.broadinstitute.org/, accessed on 15 January 2022), Varseak (https://varseak.bio/, accessed on 15 January 2022), Mutation Taster (https://www.mutationtaster.org/, accessed on 15 January 2022) and NNSplice (https://www.fruitfly.org/seq_tools/splice.html, accessed on 15 January 2022) with the default parameters.

### 2.5. Minigene Assay

To analyze how the identified variants affect splicing, the minigene assay was built as previously described [17]. We used the pcMINI-C vector that we developed previously, which contained a multicloning site as shown in Appendix A. The c.-1-2A>T variant and the c.2247-2A>G variant were located at the donor splice site of intron 2 and intron 12, respectively. Thus, exon 3 or exon 13 of *KIAA0825,* and approximately 450 bp of flanking 5′ and 3′ intronic sequences, were amplified by PCR from the controls’ genomic DNA by DNA Polymerase (PrimerSTAR MAX DNA Polymerase, R045A, TaKaRa, Kusatsu, Japan). The PCR products were cloned into a pcMINI-C vector using classical restriction and ligation methods (detailed methods seen in the Appendix A). Mutant constructs containing the c.-1-2A>T and c.2247-2A>G variants were generated by PCR using the mutant primers (detailed methods seen in the Appendix A). After the WT/mutant plasmids were directly confirmed by Sanger sequencing, both WT and mutant plasmids were transfected into HEK293T and HeLa cells. After 48 h, total RNA was extracted using the RNA extraction kit (Trizol RNAiso PLUS, 9109, TaKaRa, Kusatsu, Japan) and reverse-transcribed with the Superscript III reverse transcriptase (HifairTM 1st Strand cDNA Synthesis SuperMix for qPCR (gDNA digester plus), 11123ES70, YEASEN, Shanghai, China), and the resulting cDNA was PCR-amplified. The amplified products were analyzed on 1.5% agarose gel electrophoresis and subsequently sequenced by Sanger sequencing.

### 2.6. Reverse Transcriptase PCR Analysis

As the fetal blood sample was unavailable, peripheral blood samples from the heterozygous parents and voluntary control subjects were collected in EDTA tubes. Total RNA extraction was performed according to the manufacturer’s protocols (Trizol RNAiso PLUS, 9109, TaKaRa, Kusatsu, Japan). The quality and concentration of RNA was determined by the NanoDrop 2000 system (Thermo Fisher Scientific, Waltham, MA, USA). The HifairTM 1st Strand cDNA Synthesis SuperMix (11123ES70, YEASEN, Shanghai, China) was used to generate cDNA following the manufacturer’s protocol. The resulting cDNA was PCR-amplified using the kit (PrimerSTAR MAX DNA Polymerase; TaKaRa, Kusatsu, Japan) with primers designed to detect exon 1 to exon 5 of *KIAA0825* for the father and exon 11 to exon 15 of *KIAA0825* for the mother. Primer sets are listed in Appendix A. The amplified products were analyzed by electrophoresis on 1.5% agarose gel containing ethidium bromide and visualized by exposure to ultraviolet light. Subsequently, products were excised from the gel, purified with a commercial kit (DNA Gel Extraction Kit, 2001250, SIMGEN, Hangzhou, China) and sequenced by Sanger sequencing.

## 3. Results

### 3.1. Identification of Compound Heterozygous Splice Site Variants in KIAA0825 Related to the Disease Phenotype

SNP array analysis showed no chromosomal abnormalities or copy number variations at the whole-genome level. To further search for the potential genetic cause of the fetal abnormalities, prenatal trio-WES was performed on the fetus and both parents. The filtering criteria for trio-WES data are listed in Appendix A. After the filtering steps, 315 variants of 105 genes were identified. These genes were then related to the ‘polydactyly’ phenotype using Phenolyzer. *KIAA0825,* located on chromosome 5q15, was predicted to be the most likely candidate gene, and the loss function of KIAA0825 was responsible for postaxial polydactyly type A10(PAPA10). Two splice site variants of *KIAA0825* were identified in the fetus, NC_000005.10 (NM_001145678.3): c.2247-2A>G (rs1305545002) and c.-1-2A>T (rs1173789302), which were inherited from the mother and father, respectively. Both variants were confirmed by Sanger sequencing (Figure 2b,c) and the primers used in PCR amplification and sequencing are listed in Appendix A. Neither variant was present in the 1000 Genomes and Exome Aggregation Consortium database, but both were present in the Genome Aggregation Database with extremely low MAF (c.2247-2A>G, MAF = 1.54832× 10^−5^; c.-1-2A>T, MAF = 5.61987 × 10^−5^). 

### 3.2. Splice Effect of the KIAA0825 c.-1-2A>T Variant

In silico predictions implied that the *KIAA0825* c.-1-2A>T variant may disrupt a conserved splice acceptor upstream of the initiation codon, which would lead to the loss of initiation codon. To confirm the pathogenicity of this splice site variant and check transcription products, we conducted minigene assay and RT-PCR with the heterozygous father’s peripheral leukocytes. HEK293T and HeLa cells were transfected with wildtype or mutant minigene vectors. The wildtype constructs showed normal splicing, whereas the c.-1-2A>T variant influenced the splicing of exon 3 by disrupting its 3′ splice acceptor site. This event led to the loss of 71 nucleotides (nt) of exon 3 (r.1_70del) (Figure 3b–d). In addition, RT-PCR results from the father’s peripheral blood also revealed two aberrant alternative splicing events, including a strong skipping of exon 3 (r.1_131del) and a deletion of exon1 to exon3 (r.1_85del) (Figure 3a,c,d).

### 3.3. Splice Effect of the KIAA0825 c.2247-2A>G Variant

In silico predictions indicated that the *KIAA0825* c.2247-2A>G variant may disrupt a conserved splice acceptor of exon 13, which could result in the loss of the splice acceptor site of exon 13 and exon 13 skipping. Minigene assay revealed both exon 13 skipping (r.2247_2357del) and loss of 10 nucleotides (nt) of exon 13 (r.2247_2256del) in both HEK293T and HeLa cells compared to the wildtype control (Figure 4b–d). As the band representing the 10 nt deletion was very weak, exon 13 skipping was considered the main consequence of this variant. Moreover, the RT-PCR products of the mother who carried the heterozygous *KIAA0825* c.2247-2 A>G variant, only showed exon 13 skipping (r.2247_2357del) compared to the normal control.

## 4. Discussion

Polydactyly is a manifestation that can be either syndromic or non-syndromic, with different prognoses. For non-syndromic polydactyly patients, undergoing surgical procedures generally result in a favorable prognosis. However, for syndromic polydactyly patients, who often suffer from other severe complications, the prognoses are usually poor. In the uterus, fetal phenotypic assessment, which is mainly obtained via ultrasound and MRI, is often not comprehensive and accurate. A precise genetic diagnosis is extremely valuable for the obstetrician to evaluate the fetal prognosis, provide genetic counseling and decide on optimal treatment options.

Multiple techniques have been performed to investigate the underlying genetic bases of fetal structural anomalies. Classical karyotype and chromosome microarray analysis (CMA) can detect aneuploidy and copy number variation, which provide about 40% diagnostic yield for these fetuses [18], while prenatal WES can provide an additional 10–20% diagnostic yield and is recommended as second-tier prenatal testing for fetuses with structural anomalies [19,20,21]. In this study, SNP array analysis did not find any chromosomal abnormalities in the fetus with PAPA, while trio-WES identified compound heterozygous variants in *KIAA0825*.

The *KIAA0825* gene is located at 5q15. To date, the function of the *KIAA0825* gene has not been characterized. Only a few studies have been performed on the mouse orthologous gene *2210408I21Rik*. *2210408I21Rik* was expressed in developing limb buds from E11.5 to E15,5 and homozygous *2210408I21Rik*^tm1 (EUCOMM)Wtsi^ knock-out mice had a drastic reduction in bone mineral density [3,9,22,23,24]. Only four disease-causing variants of *KIAA0825* have been reported in patients with various PAP phenotypes in Pakistani consanguineous families [3,9,13]. In our study, we first reported two splice site variants, which were the first splice site and the fifth and sixth identified variants in the *KIAA0805* gene. The fetal clinical features in our family were generally consistent with those of previous studies [3,9,13]. Our study further validated the relationship between *KIAA0825* and PAPA10.

KIAA0825 has two isoforms: a long isoform with 1275 amino acids and a short isoform with 324 amino acids. The c.-1-2A>T variant affected both long and short isoforms. Our minigene assay showed that the c.-1-2A>T variant led to the loss of 71 nucleotides of exon 3 (r.1_70del). However, blood RT-PCR of the heterozygous father showed that the c.-1-2A>T variant created two aberrant splicing results: skipping of exon 3 (r.1_131del) and deletion of exon 1 to exon 3 (r.1_85del). The canonical initiation codon of *KIAA0825* is located at the 2nt of exon 3. Although the blood RT-PCR and minigene assay results were not consistent, both tests demonstrated that the c.-1-2A>T variant caused aberrant splicing, which led to the loss of the canonical initiation codon. The c.-1-2A>T belongs to a canonical ± 1,2 splice site. Destroying the canonical ± 1,2 splice site is often assumed to lead to a null effect and subsequently mRNA degradation through nonsense-mediated decay (NMD) without protein production [25]. However, many functional studies also showed that start loss variants could robustly reinitiate at alternate downstream ATG or non-ATG sites [26,27], likely resulting in the production of truncated proteins. Supek F. also reported that the NMD efficiency was reduced in the 5′-most 150 nt of a transcript’s coding region, gradually increasing from 5′ to 3′ in this segment [28]. Thus, based on the prediction software (https://web.expasy.org/translate/, accessed on 23 June 2022)), we speculated that the c.-1-2A>T variant would reinitiate at an in-frame ATG codon located at 336 bp downstream from the original translation initiation site (TIS), which may produce an N-terminal truncated protein lacking 112 amino acids [p.(Asn1_Glu112del)] (Appendix A). Due to the absence of a specific antibody and low *KIAA0825* expression in the peripheral blood, we cannot elucidate the exact alternate TIS of *KIAA0825*.

The c.2247-2A>G variant detected in our study only affected the long isoform. The minigene assay analysis and blood RT-PCR analysis of this variant showed that this variant produced alternative splicing products, which mainly resulted in exon 13 skipping (r.2247_2357del). This in-frame deletion of exon 13 was predicted to evade NMD and produced a truncated protein with the deletion of 37 amino acids [p.(Phe751_Thr787del)] (Appendix A). Since the fetus carries the compound heterozygous variants of *KIAA0825*, we hypothesize that both the c.-1-2A>T variant, causing a truncated protein lacking N-terminal residues, and c.2247-2A>G variant, causing a truncated protein with the in-frame deletion of 37 amino acids, contribute to the PAPA phenotype of the fetus. We also speculate that the N-terminal residues [p.(Asn1_Glu112del)] and 37 amino acids of exon 13 [p.(Phe751_Thr787del)] are very important for KIAA0825 function.

At least 50% of disease-causing variations are estimated to be splice site variants. Functional analyses of the spliced isoforms such as the minigene assay and RT-PCR can reclassify up to 75% of putative splicing variants [29]. The c.-1-2A>T and c.2247-2A>G variants belong to special types of PVS1 null variants. Thus, when we initially classified those two variants, we decreased the PVS1 criteria from very strong to supporting (c.-1-2A>T) and strong (c.2247-2A>G), then classified them as variants of unknown significance (VUS), according to the ACMG standards and guidelines for the interpretation of variations [25,28,30]. To further investigate the pathogenicity, we tested the alternative splicing of those two variants by minigene assay in both HEK293 and Hela cells and the blood RT-PCR for both heterozygous parents. After both splice isoform analyses proved that these two variants caused aberrant splicing, we reclassified them from VUSs to likely pathogenic variants. Accurate variant interpretation is essential for disease diagnosis, genetic counseling and prenatal counseling. In our study, after reaching a definitive molecular diagnosis of the fetus with PAPA, the couple decided to continue the pregnancy.

## 5. Conclusions

In conclusion, we reported two novel *KIAA0825* splice site variants in a fetus with PAPA. Using minigene assay and blood RT-PCR for both heterozygous parents, we showed that these two variants influenced splicing and potentially produced truncated proteins lacking N-terminal residues [p.(Asn1_Glu112del)] and 37 amino acids of exon 13 [p.(Phe751_Thr787del)]. Prenatal trio-WES and subsequent functional studies can identify the underlying etiology of the fetus with PAPA, help to evaluate fetal prognosis, provide genetic counseling related to recurrence risk and facilitate future preimplantation genetic diagnosis and prenatal diagnosis.

## Figures and Tables

**Figure 1 genes-13-01230-f001:**
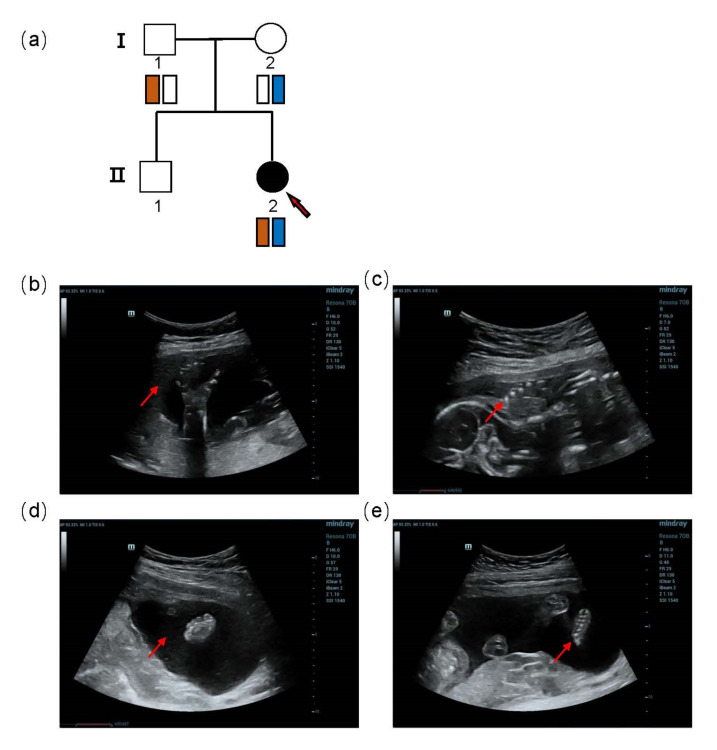
Family pedigree and Ultrasound imaging data of the fetus showed PAPA. (**a**) A Pedigree of this family with segregating PAP in an autosomal recessive manner. Circles and squares represent females and males, respectively. Clear symbols represent unaffected members, while filled symbols show affected members. (**b**,**c**) Ultrasound imaging for right hand (**b**) and left hand (**c**) displaying PAPA. (**d**,**e**) Ultrasound imaging for right foot (**d**) and left foot (**e**) exhibiting PAPA.

**Figure 2 genes-13-01230-f002:**
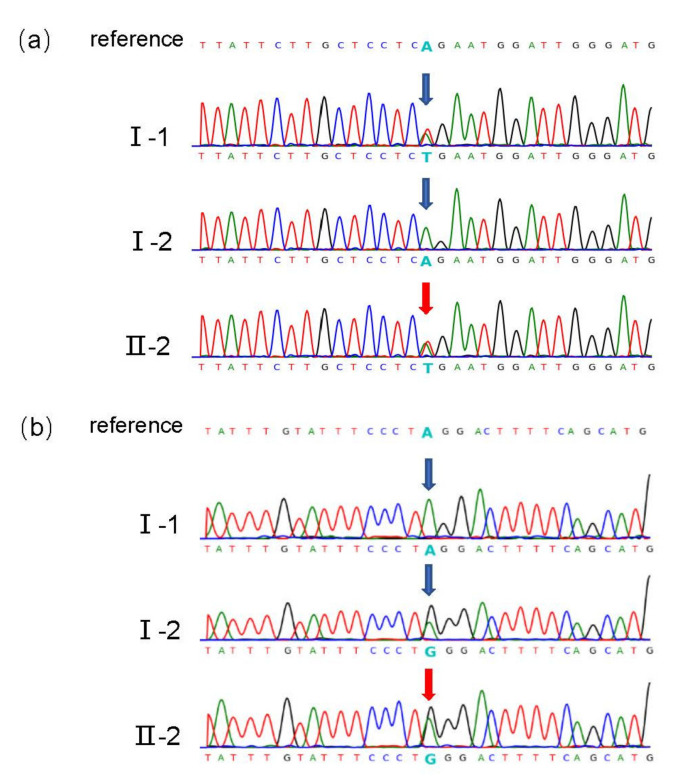
Sanger sequencing chromatogram of *KIAA0825* variants in the family. (**a**) The variant c.-1-2A>T in *KIAA0825* is identified in II-2 (proband) and I-1 (father). Arrows represent the variant. (**b**) The variant c.2247-2A>G in *KIAA0825* is identified in II-2 (proband) and I-2 (mother). Arrows represent the variant.

**Figure 3 genes-13-01230-f003:**
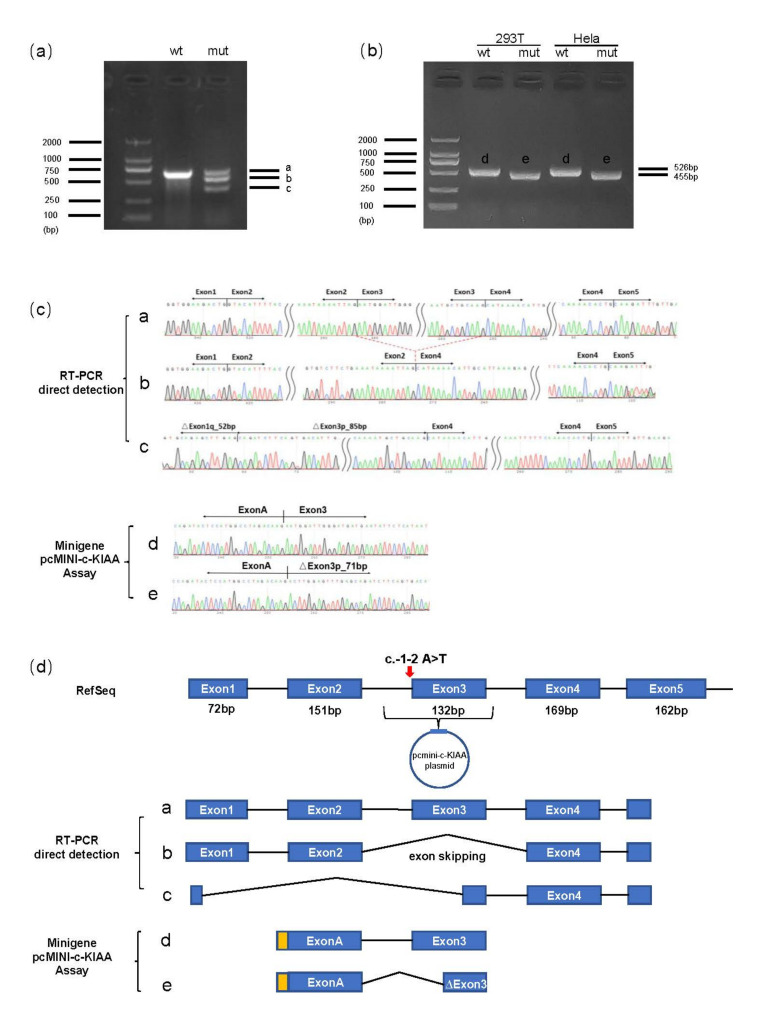
Blood RT-PCR and minigene assay analysis of the *KIAA0825* c.-1-2A>T. (**a**) Electrophoresis results of the RT-PCR products with the father’s peripheral leukocytes; the band amplified from the control (wt) is labeled as a and the bands amplified from the father (mut) are labeled as a, b and c. (**b**) RT-PCR electrophoresis results of the minigene assay; bands from WT and the c.-1-2A>T constructs were labeled as d and e, respectively, in both HeLa and 293T cells. (**c**) The corresponding Sanger sequencing results of excised bands of a, b, c from the blood RT-PCR and excised bands of d, e from the minigene assay. (**d**) The diagram of alternative splicing events observed from the sanger sequence in the blood RT-PCR and minigene assay. The alternative splicing events from the father (mut) are labeled as a, b and c. The alternative splicing events from WT and the c.-1-2A>T constructs in minigene were labeled as d and e. Red arrow indicates the variant location.

**Figure 4 genes-13-01230-f004:**
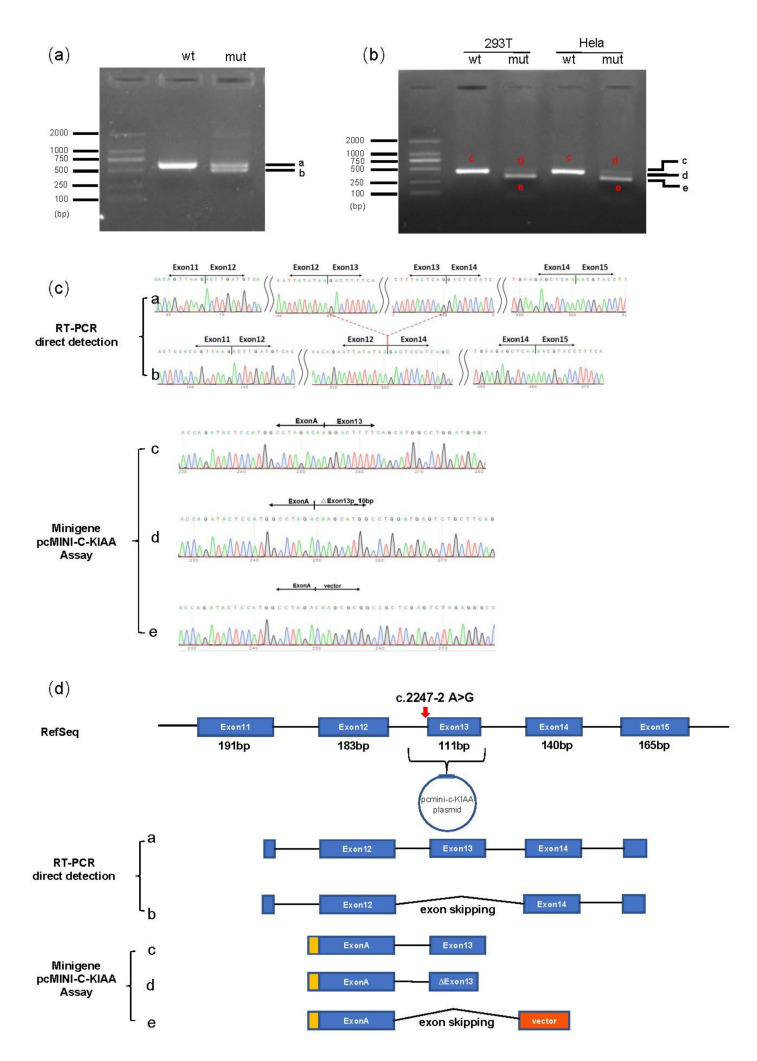
Blood RT-PCR and minigene assay analysis of the *KIAA0825* c.2247-2A>G variant: (**a**) Electrophoresis results of the RT-PCR products with the mother’s peripheral leukocytes, the band amplified from the normal control (wt) was labeled as a and the bands amplified from the mother (mut) were named as a and b. (**b**) RT-PCR electrophoresis results of the minigene assay. Bands from the WT and c.2247-2A>G constructs were labeled as c, d and e, respectively, both in HeLa and 293T cells. (**c**) The corresponding Sanger sequencing results of excised bands of a, b from the blood RT-PCR and excised bands of c, d, e from the minigene assay. (**d**) The diagram of alternative splicing events observed in the blood RT-PCR and minigene assay. The alternative splicing events from the mather (mut) are labeled as a and b. The alternative splicing events from WT and the c.2247-2A>G constructs in minigene were labeled as c, d and e. Red arrow indicates the variant location.

## Data Availability

The raw sequence data reported in this paper have been deposited in the Genome Sequence Archive in National Genomics Data Center, China National Center for Bioinformation/Beijing Institute of Genomics, Chinese Academy of Sciences (https://bigd.big.ac.cn/, accessed on 15 June 2022). After the publication of the study findings, the data will be available for others to request. The corresponding author will provide the accession number of the database once the data are approved to be shared with others. A proposal with the description of study objectives will be needed for the evaluation of the reasonability to request the data. The corresponding author and the Ethics Committee of Maternal and Child Health Hospital of Hubei Province, Tongji Medical College, Huazhong University of Science and Technology, Wuhan, China will make a decision based on these materials.

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
