# Peer review of "Prenatal Detection of Novel Compound Heterozygous Splice Site Variants of the KIAA0825 Gene in a Fetus with Postaxial Polydactyly Type A"

_genes, 2022, doi:10.3390/genes13071230_

Round 1

Reviewer 1 Report

Paper for review

Yanyi Yao et al. submitted a manuscript entitled:Case Report: Prenatal detection of novel biallelic splice site variants of the KIAA0825 gene in a fetus with postaxial Polydactyly”

They report the 4rd report in literature. The clinical and molecular data very are very interesting and could be of interest to the readers of the Frontiers in Genetics. However, there are minor comments that should be seriously addressed.

General comments:

*- The reference sequence used should be described properly, the version number (e.g. NG_0123456. 3; NP_5462.2) is missing. ** Please note that since the reference sequence given (NM_) does not contain intronic sequences a genomic reference sequence should be given as well (e.g. NG_0123456.3).

** English language editing is required. Upload English editing certificate.

Abstract:

Well written

Introduction

++ Its not FAM98A, its FAM92A [https://pubmed.ncbi.nlm.nih.gov/30395363/]. Page 2 line 52.

Methods and results

*Spacing mistakes, kindly correct.

+Better to show the filtration steps used to screen the variants using a flow sheet.

++ if exon 3 is skipped, what will be the (p.?)?

Discussion

Written in detail.

Correct the mutation.

Figures

Improve the quality and font should be same. Poor quality figures.

Reviewer 2 Report

Review comment

In the present manuscript, it suggests that the novel compound heterozygous splice site variants of the KIAA0825 gene, the responsible gene of the postaxial polydactyly type A (PAPA), may cause the PAPA disease. The authors found two putative heterozygous splicing site variants in a fetus with PAPA and confirmed that the variants cause abnormal splicing variant by RT-PCR and minigene assay. Also, two variants are compound heterozygous variants by trio-WES and trio-Sanger sequencing. I recommend some minor revisions. 

1. RT-PCR analysis showed that in vivo KIAA0825 transcripts owing to use of the peripheral leukocytes. However, both minigene analysis and RT-PCR analysis are in vitro analysis. The authors should correct in vivo RT-PCR analysis.  

2. Two variants have registered in dbSNP which is the famous SNP database. It is helpful to add the reference number. The variant (NM_001145678.1:c.-1-2A>T) is rs1173789302, the variant (NM_001145678.1:c.2247-2A>T) is rs1305545002.

3. The variant (NM_001145678.1:c.-1-2A>T) may produce a N-terminal truncated protein due to exon skipping. The authors should be add the predictable protein sequence such as p.(Phe751_Thr787del).

4. In abstract, PAPA is abbreviation. It is better not to use abbreviations from the beginning. 

5. In Materials and Methods, although the title is Minigene reporter assay, the rest are the minigene assay. The authors should unify the description to either.

6. In line 125, the authors should correct Retro-transcribed to reverse transcribed.

7. In line 143, the authors should correct excited to excised.

8. In line 155, Figure 2B-C are from sanger sequencing. However, it is written as the data of the trio-WES in the manuscript.

9. In line 199, it is not the shear band but the excised band.

10. The lowercase latters in Figures. On the other hand, uppercase latters in the manuscript.

Author Response

This manuscript is a resubmission of an earlier submission. The following is a list of the peer review reports and author responses from that submission.